# Tuberculosis and Non-Communicable Disease Multimorbidity: An Analysis of the World Health Survey in 48 Low- and Middle-Income Countries

**DOI:** 10.3390/ijerph18052439

**Published:** 2021-03-02

**Authors:** Brendon Stubbs, Kamran Siddiqi, Helen Elsey, Najma Siddiqi, Ruimin Ma, Eugenia Romano, Sameen Siddiqi, Ai Koyanagi

**Affiliations:** 1Physiotherapy Department, South London and Maudsley NHS Foundation Trust, Denmark Hill, London SE5 8AZ, UK; 2Department of Psychological Medicine, Institute of Psychiatry, Psychology and Neuroscience (IoPPN), King’s College London, London SE5 8AB, UK; ruimin.1.ma@kcl.ac.uk (R.M.); eugenia.romano@kcl.ac.uk (E.R.); 3Department of Health Sciences, The University of York, York YO10 5DD, UK; kamran.siddiqi@york.ac.uk (K.S.); helen.elsey@york.ac.uk (H.E.); najma.siddiqi@york.ac.uk (N.S.); 4Hull York Medical School, The University of York, York YO10 5DD, UK; 5Department of Community Health Sciences, Aga Khan University, Karachi 74800, Pakistan; sameen.siddiqi@aku.edu; 6Research and Development Unit, Parc Sanitari Sant Joan de Déu, CIBERSAM, 08830 Barcelona, Spain; a.koyanagi@pssjd.org; 7ICREA, Pg. Lluis Companys 23, 08010 Barcelona, Spain

**Keywords:** tuberculosis, non-communicable diseases, comorbidities, low- and middle-income countries, multimorbidity

## Abstract

Tuberculosis (TB) is a leading cause of mortality in low- and middle-income countries (LMICs). TB multimorbidity [TB and ≥1 non-communicable diseases (NCDs)] is common, but studies are sparse. Cross-sectional, community-based data including adults from 21 low-income countries and 27 middle-income countries were utilized from the World Health Survey. Associations between 9 NCDs and TB were assessed with multivariable logistic regression analysis. Years lived with disability (YLDs) were calculated using disability weights provided by the 2017 Global Burden of Disease Study. Eight out of 9 NCDs (all except visual impairment) were associated with TB (odds ratio (OR) ranging from 1.38–4.0). Prevalence of self-reported TB increased linearly with increasing numbers of NCDs. Compared to those with no NCDs, those who had 1, 2, 3, 4, and ≥5 NCDs had 2.61 (95% confidence interval (CI) = 2.14–3.22), 4.71 (95%CI = 3.67–6.11), 6.96 (95%CI = 4.95–9.87), 10.59 (95%CI = 7.10–15.80), and 19.89 (95%CI = 11.13–35.52) times higher odds for TB. Among those with TB, the most prevalent combinations of NCDs were angina and depression, followed by angina and arthritis. For people with TB, the YLDs were three times higher than in people without multimorbidity or TB, and a third of the YLDs were attributable to NCDs. Urgent research to understand, prevent and manage NCDs in people with TB in LMICs is needed.

## 1. Introduction

Tuberculosis (TB) is the leading cause of deaths from a single infectious disease globally [1]. There were approximately 10 million TB cases worldwide, and 1.5 million people died from TB in 2018 [1]. People from low- and middle-income countries (LMICs) are at a high risk of developing TB [2,3], with over 95% of TB cases and deaths occurring in LMICs [1]. Significant factors associated with increased risk of TB have been widely reported, including malnutrition, diabetes [4], poverty, overcrowding [5], alcoholism, smoking [6], human immunodeficiency virus (HIV) and chronic pulmonary disease [7].

As a long-term infectious disease, TB may lead to the breakdown of immune surveillance [7], increasing one’s susceptibility to other conditions such as non-communicable diseases (NCDs) [6,8] (e.g., cancer, diabetes, respiratory and cardiovascular diseases), which together contribute to two thirds of the global mortality [9]. Moreover, there is evidence of an increased prevalence of mental health problems in those diagnosed with TB [10,11]. The relationship between TB and chronic physical conditions may be bidirectional. Several health conditions, such as diabetes [12,13,14] and chronic lung diseases including chronic obstructive pulmonary disease and silicosis [6], have been previously reported as important risk factors for TB.

The co-occurrence of two or more chronic diseases in one individual at one point of his/her life is typically referred to as multimorbidity [15,16]. Multimorbidity is becoming a primary concern in our global healthcare system, especially for LMICs where there is a need to consider economic restraints when planning levels of care [17]. There is evidence that multimorbidity may occur in 1 out of 4 adults across LMICs [18], and that its prevalence may be increasing due to increasing life expectancy in these settings [19]. The determinants of multimorbidity include socioeconomic factors (e.g., unemployment, deprivation level, economic restrains), low accessibility to healthcare [20], and health risk behaviors (e.g., low physical activity, poor diet) [21,22]. All these determinants are very evident and alarming in LMICs [23].

In countries where TB is endemic [1], a concerning rapid increase in NCDs has been reported, especially in sub-Saharan Africa [3,24] and in South Asian regions [25], and particularly for those NCDs associated with TB. Despite this, evidence on NCD multimorbidity in people with TB in these areas remains scarce, with numerous gaps and limitations in the literature. First, most research considering TB multimorbidity—presence of TB alongside one or more chronic conditions—has included small-scale studies [26], been restricted to a single country [4,27,28], or only considered one comorbid chronic condition [29,30]. Second, limited information is available on the actual patterns and risks of multimorbidity compared to the general population and on the burden of disease associated with TB multimorbidity [31]. Nonetheless, some preliminary research has suggested that TB multimorbidity is associated with increased use of healthcare services [32,33], increased symptom complexity [34], high premature mortality risk [35,36], increased disabilities [37,38], poor quality of life [38,39,40], frequent hospitalization [41,42], and high healthcare expenses [6]. Despite the negative consequences associated with TB multimorbidity, TB services tend to be single focus programs that rarely screen for NCDs, and NCD care itself is also limited and rarely provided, especially in LMICs. Therefore, given the aforementioned knowledge gap, the aim of this study was to utilize a large multinational cross-sectional study to assess the pattern and burden of multimorbidity in adults with TB and NCDs across 48 LMICs.

## 2. Materials and Methods

### 2.1. The Survey

The World Health Survey (WHS) was a cross-sectional survey carried out in 70 countries in 2002–2004 (data freely available for analysis). Survey details are available elsewhere [43]. Briefly, single-stage random sampling and stratified multi-stage random cluster sampling was conducted in 10 and 60 countries, respectively. Eligible participants were those with a valid home address and aged ≥18 years. One individual was randomly chosen from each household with the use of Kish Grid tables [44]. The questionnaire was subject to standard translation procedures to ensure comparability between countries. The questionnaire was administered face-to-face by trained interviewers. The overall individual response rate was 98.5% [45]. To adjust for non-response, sampling weights were generated using the population distribution as reported by the United Nations Statistical Division [46]. Ethical approval for the survey was provided by ethical boards at each study site. All participants gave their informed consent.

### 2.2. Variables

#### 2.2.1. Tuberculosis

TB case definition was based on self-reported symptoms of active disease in the past 12 months. The survey procedures did not include mycobacterial culture or sputum smear examinations. Specifically, as in previous WHS publications [11,47,48,49], those who had both (a) a cough that lasted for three weeks or longer, and (b) blood in phlegm (or coughed up blood) were considered to have active TB. Previous studies have shown that the presence of these typical symptoms is likely to have sensitivity and specificity of 65–70% and 55–75% to detect TB, respectively [47].

#### 2.2.2. Non-Communicable Diseases (NCDs)

A total of nine NCDs (angina, arthritis, asthma, diabetes, depression, edentulism, hearing problem, schizophrenia, and visual impairment) were assessed in the WHS. Arthritis, asthma, and diabetes were based on self-reported lifetime diagnosis. For angina, in addition to a self-reported diagnosis, a symptom-based diagnosis based on the Rose questionnaire was used [50]. The participant was considered to have hearing problems if the interviewer observed this condition at the end of the survey. Depression was defined using the Diagnostic and Statistical Manual of Mental Disorders (4th Edition; DSM-IV) algorithm and was based on duration and persistence of depressive symptoms in the past 12 months using the same algorithms as in previous WHS publications [51,52]. Edentulism was defined as an affirmative response to the question “Have you lost all your natural teeth?” Participants were asked whether they had ever been diagnosed with schizophrenia or psychosis. Those who answered affirmatively were considered to have schizophrenia. While this question did not refer specifically to schizophrenia, for the sake of brevity, we use the term ‘schizophrenia’ throughout the text. Visual impairment was defined as having extreme difficulty in seeing and recognizing a person that the participant knows across the road (i.e., from a distance about 20 m) [53]. A validity study showed that this response likely corresponds to WHO definitions of visual impairment (20/60 or 0.48 The logarithm of the minimal angle of resolution, logMAR) [53]. We calculated the total number of these NCDs while allowing for one missing variable in order to retain a larger sample size. TB multimorbidity was defined as having TB with one or more NCDs [8].

#### 2.2.3. Control Variables

The selection of the control variables was based on past literature and included age, sex, highest education achieved (no formal education, primary education, secondary or high school completed, or tertiary education completed), wealth, current smoking, and heavy episodic drinking [8,54]. Country-wise wealth quintiles were created using principal component analysis based on 15–20 assets including country-specific items for some countries. Heavy episodic drinking was defined as having consumed ≥4 (female) or ≥5 (male) standard alcoholic beverages on ≥2 days in the past 7 days [55].

### 2.3. Statistical Analysis

Publicly available data of the WHS included 69 countries. The data were nationally representative for all countries with the exception of China, Comoros, the Republic of Congo, Ivory Coast, India, and Russia. We excluded 10 countries as they lacked sampling information. A further 10 high-income countries were excluded as the focus of the study was on LMICs. Finally, Turkey was excluded due to lack of information on education and diabetes. Thus, a total of 48 countries, of which 21 (*n* = 105,286) and 27 (*n* = 137,666) were low-income and middle-income countries, respectively, at the time of the survey (2003) according to the World Bank classification, were included in the final sample. Information on the individual countries is provided in Table A1.

The statistical analysis was performed with Stata 14.1 (Stata Corp LP, College station, TX, USA). The difference in sample characteristics between those with and without TB was tested by Chi-squared tests and Student’s *t*-tests for categorical and continuous variables, respectively. In patients with TB, tetrachoric correlations between each pair of NCDs and the prevalence of their co-occurrence were calculated. We also estimated and compared years lived with disability (YLDs) in both people with and without TB. YLDs were based on the prevalence of and the disability weights [56] assigned to the nine NCDs and TB. Limited by the cross-sectional data, we assumed that people lived in the same health status for one year. Using the overall sample, the association of each NCD, and number of NCDs (exposures) with TB (outcome) was assessed with multivariable logistic regression analysis. The association between NCD multimorbidity and TB was also assessed by sex, age group, and country-income level. All regression analyses were adjusted for age, sex, wealth, education, smoking, heavy drinking, and country with the exception of the sex-stratified analysis, which was not adjusted for sex. Adjustment for country was done by including dummy variables in the models, as in previous WHS publications [48,57]. All variables were included in the models as categorical variables with the exception of age (continuous variable). In order to assess the influence of multicollinearity, we calculated the variance inflation factor (VIF) value for each independent variable. The highest VIF was 1.43, which is much lower than the commonly used cut-off of 10 [58], indicating that multicollinearity was unlikely to be a problem in our analyses. The sample weighting and the complex study design were taken into account in all analyses. Results from the logistic regression are presented as odds ratios (ORs) with 95% confidence intervals (CIs). The level of statistical significance was set at *p* < 0.05.

## 3. Results

The final sample consisted of 242,952 adults aged ≥18 years, mean (SD) age 38.4 (16.0) years and 50.8% were female (Table 1). The overall prevalence of TB was 1.7%, while the prevalence of TB multimorbidity (i.e., TB and at least one NCD) was 1.12%. The prevalence of male sex, lower levels of wealth and education, smoking, and all NCDs as well as greater number of NCDs was significantly higher among those with TB. The prevalence of TB increased linearly with increasing number of NCDs (Figure A1). Specifically, the prevalence was 0.8% among those with no NCDs but increased to 14.7% among those with ≥5 NCDs. The associations between the individual NCDs and TB estimated by multivariable logistic regression are shown in Figure 1. Except for visual impairment, all NCDs were significantly associated with TB with particularly strong associations (i.e., OR ≥3) observed for asthma, angina, and depression. In terms of the adjusted association between the number of NCDs and TB, there was a linear increase in the ORs with increasing number of NCDs (Figure 2). Specifically, compared to those with no NCDs, the OR for TB was 2.61, 4.71, 6.96, 10.59, and 19.89 for 1, 2, 3, 4, and ≥5 NCDs, respectively.

The prevalence of one or more of these NCDs was 68.8% in people with TB, and 34.4% in people without TB. Among those with TB, the strongest correlations in terms of the pairs of NCDs were observed for depression and schizophrenia, followed by arthritis and angina (Figure 3). The most prevalent pairs of concurring NCDs among people with TB were angina and depression (15.3%), followed by arthritis and asthma (14.6%), asthma and angina (10.7%), and depression and arthritis (9.7%) (Figure 4). Figure 5 highlights that a third of YLDs in people with TB could be attributable to NCDs including depression, angina, schizophrenia, asthma, arthritis and diabetes. The YLDs attributable to NCDs were three times higher in TB than the YLDs attributable to NCDs in participants without TB. Moreover, in people with TB, just over 55% of total YLDs were in patients with 2 or more NCDs, 24% in patients with one NCD and 21% in patients with no NCD.

## 4. Discussion

To the best of our knowledge, the current study is the first representative multinational LMIC study to investigate patterns of NCDs and YLDs associated with TB multimorbidity. Almost all NCDs were significantly associated with TB with high odds observed for angina, asthma and depression (OR ≥ 3.0). Moreover, we found a concerning linear increase in the odds of TB with increasing number of NCDs. Furthermore, our data found that YLDs attributable to NCDs were three times higher in people with TB than in those without TB. Given the high prevalence of TB and multimorbidity, our results shed important new light on the importance of understanding and preventing NCD comorbidity in TB.

While our study consisted of a relatively young adult sample (mean age = 38.4), the prevalence of any NCD multimorbidity in our sample was relatively high (i.e., 13%). Advancing age has been associated with a higher multimorbidity rate, given the development process from most biological pathways to chronic health conditions can be long [34]. The prevalence of multimorbidity can be as high as 72% in older adults [39]. Reis-Santos and colleagues [27] discovered that people with TB aged 60 and over in Brazil were 44 times more likely to develop TB multimorbidity than their younger counterparts.

Our results suggest that male sex, lower levels of wealth and education, smoking, prevalence of NCDs and associated multimorbidity were significantly higher among people with TB than those without a TB diagnosis. Our results are consistent with previous findings: male sex [26,59] and smoking [6,7,60] are two important risk factors for the elevated incidence of TB, and it has also been suggested that people with low educational attainment and poor income are at a higher risk of developing TB [61,62]. The cost of healthcare is a particularly relevant factor in LMICs, where the high healthcare cost relating to NCDs may lead to increased financial restraints and financial burden for accessing preventive health programs and treatment in low-income households [63,64,65].

Our results showed that the odds for TB increased linearly with the number of NCDs, and that people with TB also reported three times the amount of YLDs due to NCDs compared to those without TB. This is important, considering that a report from 2004 [66] did not report TB among the conditions impacting YLDs in LMICs, so more focus on TB is recommended for future investigations. Our results on the coexistence of NCDs and TB are not surprising, considering that both can impact the organisms on several levels. There is some existing evidence of the role of NCDs in the development of TB in the literature, but this has typically been limited to a single country, non-representative samples or considered a single or limited number of NCDs at any one time. A study of a Korean population, for example, has found that both NCD and their risk factors can be the determinants for the levels of registered TB in the local population [67]. However, we need to consider that the association between TB and NCDs might be bidirectional. For example, diabetes (a risk factor for TB [68]), can affect TB treatment, and increase the chance of relapse and fatality rate [13]. At the same time, infection due to TB could affect glucose tolerance [69] or glycaemia, hence posing a risk for aggravation in diabetes [70,71]. In terms of the association between depression and TB, while TB could worsen depression due to its effect on quality of life [72] and social stigma [73], depression itself may also affect our immune system [74] and be an actual risk factor for TB [75]. Depression can also further aggravate TB, since it is a risk factor for lack of compliance in medical treatment [76], and certain anti-depressants might interfere with anti-TB agents [77]. Furthermore, a study on people with pulmonary TB has also reported that comorbidity with depression can lead to multi-drug resistance [78], possibly due to factors such as non-compliance to treatment. Given the cross-sectional nature of the World Health Survey, a full discussion of the causal association and direction of the relationship between NCD and TB is beyond our scope. Nevertheless, our results were sufficient to assess the extent of NCD multimorbidity in TB.

Our study also found that the most prevalent pairs of co-occurring NCDs in people with TB were depression and angina, arthritis and angina, asthma and angina, and depression and arthritis. Angina showed several interesting associations in terms of multimorbidity according to our results. There is evidence on how patients with early psoriatic arthritis report higher rates of angina compared to the general population [79], while the co-occurrence of angina and asthma seems to be common in males according to a study by Cazzola and colleagues [80]. The association of depression with both angina and arthritis could be due to the impact of such diseases on normal functioning. In fact, there is evidence suggesting an association of angina symptoms with anxiety, depression, and impaired physical functioning [81]. A meta-analysis by Matcham and colleagues [82] reports a high prevalence of depression in rheumatoid arthritis, as well as an association with poorer outcomes related to arthritis. Our findings have important implications in informing policy change and clinical practice. The key message is that when managing TB patients, due to the co-occurrence of TB with NCDs, some specific NCD multimorbidity clusters (e.g., angina and depression) deserve early detection and specific care. Frequent screening of NCDs should be provided in routine TB care. Furthermore, considering the burden and frequency of the co-occurrence of depression with NCDs [83] and with TB [60], our results suggest the need for a strategy for management that can take all these factors into consideration. Finally, continuous education regarding the co-occurrence of TB and specific NCDs or NCD multimorbidity for healthcare providers may lead to early detection of TB multimorbidity and better patient care.

Altogether, our findings are consistent with the so called “syndemic theory”, which addresses the co-occurring diseases that afflict low-income populations from multiple perspectives [84]. According to this model, co-occurring conditions share similar pathophysiology, predisposed by biological factors, and affect people not only through somatic symptoms, but also by undermining their daily activities and personal and social experiences in society. All of this is influenced by structural and societal inequalities, which could expose people to risk factors for certain comorbidities, and by cultural factors, which could impact the response to illness and disabilities. Considering the syndemic perspective, our study provides further evidence on the co-occurrence of physical and mental conditions, hence underlining the increasing need to address health inequalities in LMICs. Multimorbidity comes with great symptom complexity which may influence time spent in the hospital and healthcare costs [34]. This is concerning given that in LMICs, there is a lack of an integrated health care system, high costs in accessing healthcare [85], and a need to plan all levels of care with the consideration of economic burden [17]. A sustainable healthcare education [3] and an integrated approach [34] should hence be recommended for health staff in these areas.

Whilst our study has substantially contributed to advancing the field and includes a representative multinational sample across 48 LMICs, some limitations should be considered. First, this study is cross-sectional in nature, hence, the causality and temporality between TB and NCDs could not be inferred. Future research with a prospective cohort design will provide rigorous evidence of the relationship between TB and NCDs. Second, the utilization of self-reported measures for NCDs and mental health diagnoses may have led to recall bias or underdiagnosis in our sample, especially in older subjects. On a similar note, the diagnosis of TB was also based on self-reported typical symptoms; therefore, there was also the possibility for misclassification. Whilst there are multiple other studies that have been published using this data in other populations [11,17,21], this consideration should be taken into account. Furthermore, distinction between drug-sensitive and multidrug-resistant TB could not be addressed. Therefore, future studies should include different types of TB and may wish to consider the inclusion of more objective diagnostic methods for physical and mental health conditions, such as patients’ hospital medical records. Third, our study included nine physical conditions but lacked information on some other important NCDs such as chronic obstructive pulmonary disease (COPD) and bronchiectasis. Therefore, the prevalence of multimorbidity could be higher if more conditions were included. Fourth, the prevalence of diabetes was lower than recent population prevalence figures in many of these countries [86], suggesting that the prevalence of diabetes in people with TB might have also gone up remarkably in recent years, or that diabetes might have been under diagnosed before. Fifth, the LMICs included in the current study were categorized based on the World Bank classification at the time of the survey (2003). Therefore, the demographics and epidemiology of many included countries are likely to have changed over the last 20 years. New surveys should be carried out and provide the most up-to-date evidence in order to inform the relationship between TB and NCDs in LMICs. Finally, our study did not include at-risk samples, such as homeless individuals or prisoners. The prevalence of chronic health conditions (e.g., infections, mental health problems) and subsequent all-cause mortality tends to be higher among marginalized populations [87,88]. Future research should thus consider these vulnerable and marginalized populations. Altogether, our results and limitations suggest the need to further explore and address the co-occurrence of NCDs and TB considering different epidemiologic, demographic, and symptomatic profiles, in order to better tailor prevention and management strategies in those countries where the risk for such comorbidities is higher, and the availability of resources is lower.

## 5. Conclusions

In summary, our data utilizing information from 48 LMICs suggests that TB is consistently associated with multiple NCDs including key physical and mental health conditions. Of concern, we identified that the YLDs were three times higher in people with TB versus people without multimorbidity or TB. Clearly, there is an urgent need for further research to understand common clusters and develop effective interventions to prevent and treat people with TB multimorbidity and reduce their associated impact in LMICs.

## Figures and Tables

**Figure 1 ijerph-18-02439-f001:**
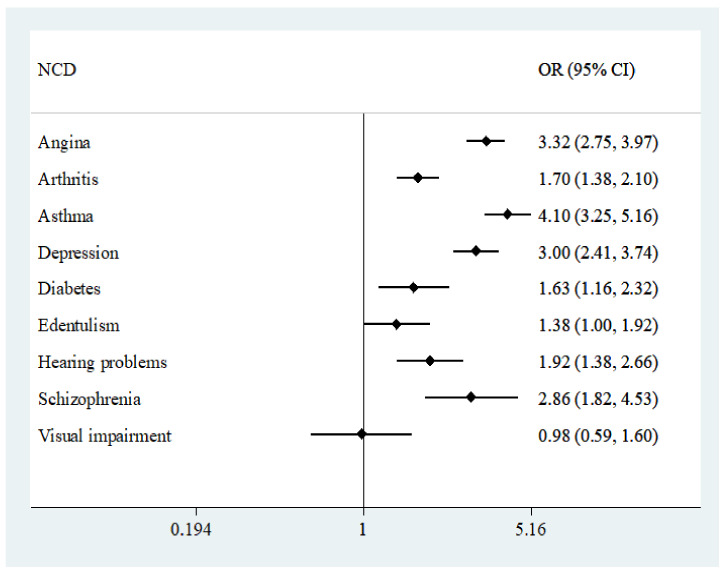
Association between non-communicable diseases and tuberculosis estimated. by multivariable logistic regression. Abbreviation: NCD Non-communicable disease; OR Odds ratio; CI Confidence interval. Models are adjusted for age, sex, wealth, education, smoking, heavy drinking, and country.

**Figure 2 ijerph-18-02439-f002:**
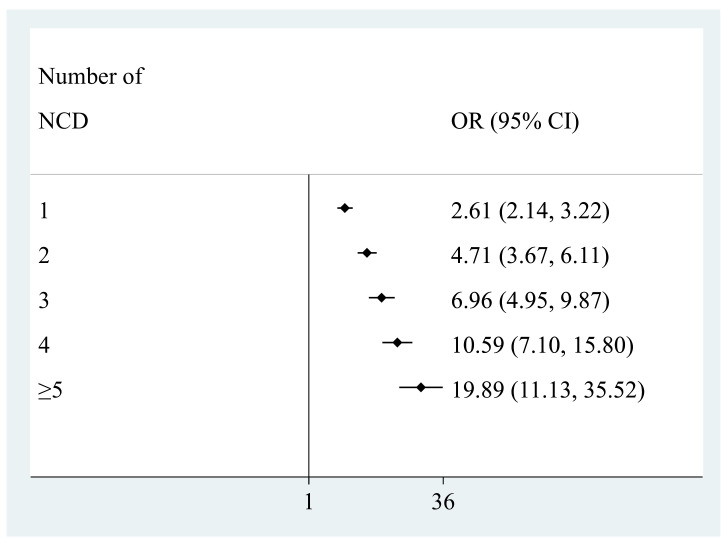
Association between number of non-communicable diseases and tuberculosis, estimated by multivariable logistic regression. Abbreviation: NCD Non-communicable disease; OR Odds ratio; CI Confidence interval. Reference category is no non-communicable disease. Model is adjusted for age, sex, wealth, education, smoking, heavy drinking, and country.

**Figure 3 ijerph-18-02439-f003:**
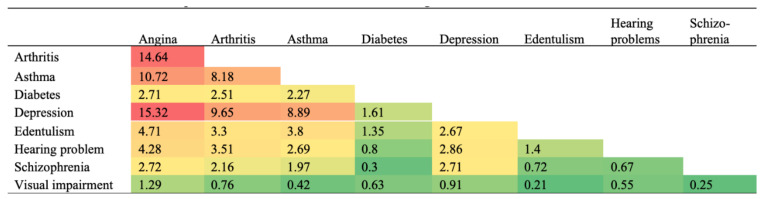
Prevalence of each pair of non-communicable diseases among individuals with tuberculosis. Numbers are presented as %.

**Figure 4 ijerph-18-02439-f004:**
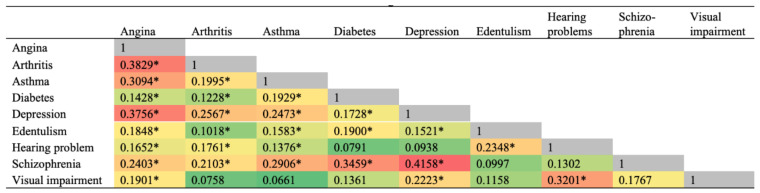
Tetrachoric correlations of non-communicable diseases among individuals with tuberculosis. * *p* < 0.05.

**Figure 5 ijerph-18-02439-f005:**
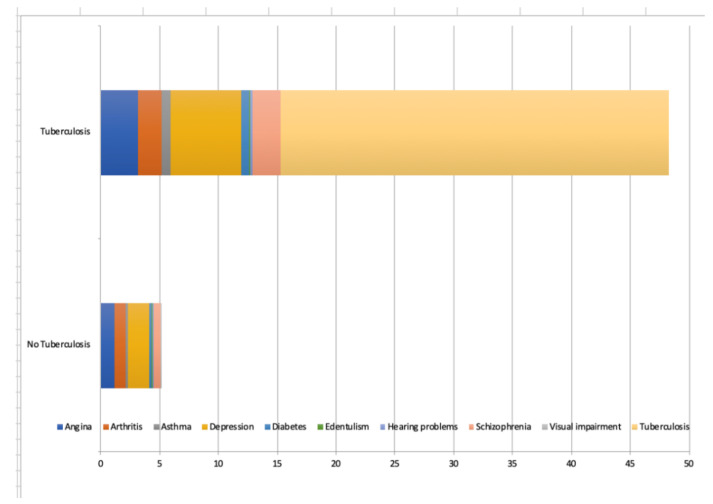
The cumulative total of years lived with disability in a cohort of people with and without tuberculosis.

**Table 1 ijerph-18-02439-t001:** Sample characteristics.

			Tuberculosis		
Characteristic		Overall	No	Yes	*p*-Value ^a^
Age (years)	Mean (SD)	38.4 (16.0)	35.5 (16.0)	43.4 (17.1)	<0.001
Sex	Male	49.2	49.1	54.0	0.010
	Female	50.8	50.9	46.0	
Wealth	Poorest	20.5	20.3	31.7	<0.001
	Poorer	20.0	19.9	22.6	
	Middle	20.0	20.0	17.7	
	Richer	19.8	19.9	16.6	
	Richest	19.8	19.9	11.4	
Education	No formal	25.2	24.9	43.9	<0.001
	Primary	31.8	31.7	34.2	
	Secondary	33.1	33.4	19.4	
	Tertiary	9.9	10.0	2.5	
Smoking	No	72.8	72.9	63.0	<0.001
	Yes	27.2	27.1	37.0	
Heavy drinking	No	97.9	98.0	97.6	0.354
	Yes	2.1	2.0	2.4	
Angina	No	85.4	85.8	60.3	<0.001
	Yes	14.6	14.2	39.7	
Arthritis	No	87.1	87.3	74.5	<0.001
	Yes	12.9	12.7	25.5	
Asthma	No	94.6	94.9	78.2	<0.001
	Yes	5.4	5.1	21.8	
Diabetes	No	96.9	96.9	93.4	<0.001
	Yes	3.1	3.1	6.6	
Depression	No	92.9	93.2	75.8	<0.001
	Yes	7.1	6.8	24.2	
Edentulism	No	93.7	93.8	89.5	<0.001
	Yes	6.3	6.2	10.5	
Hearing problems	No	96.5	96.6	91.6	<0.001
	Yes	3.5	3.4	8.4	
Schizophrenia	No	98.9	99.0	95.5	<0.001
	Yes	1.1	1.0	4.5	
Visual impairment	No	98.6	98.6	97.8	0.030
	Yes	1.4	1.4	2.2	
No. of NCD	0	65.6	65.6	31.2	<0.001
	1	21.8	21.8	29.2	
	2	8.4	8.4	20.4	
	3	3.0	3.0	11.3	
	4	0.9	0.9	4.6	
	≥5	0.3	0.3	3.3	

Abbreviation: SD Standard deviation; NCD Non-communicable disease. Data are % unless otherwise stated. ^a^
*p*-value was based on Chi-squared tests for categorical variables and Student’s *t*-tests for continuous variables (age).

## Data Availability

The datasets generated and/or analyzed during the current study are publicly available in the World Health Survey repository.

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
