# Peer review of "Tuberculosis and Non-Communicable Disease Multimorbidity: An Analysis of the World Health Survey in 48 Low- and Middle-Income Countries"

_ijerph, 2021, doi:10.3390/ijerph18052439_

Round 1

Reviewer 1 Report

The authors aimed to explore the pattern and burden of multimorbidity in adults with TB and NCDs in 48 LMICs. Considering the established association between TB and NCDs and its scarce data in LMICs, this study gives an added value to the field. However, I would like to raise some concerns for publication.

  1. Study limitations: The discussion section indicates several limitations. However, the most important limitation would be the study design. A cross-sectional study does not determine the causal relationship between factors. We do not know how these factors are correlated through this study design. This should be indicated in the discussion section. In addition, it would be better to mention how to have tried to address all other limitations in the manuscript.
  2. Study implications: What are the implications of the study? How can we apply these findings in the field? Which policy should be adopted to address these findings? The manuscript does not clearly mention these.
  3. Possible multicollinearity: In the multivariable logistic regression analysis, several variables are included for adjustment including income and wealth. In particular, I am concerned about possible multicollinearity between income level and education status. These two factors are usually correlated so that if two factors are included in a multivariable analysis, the result might be biased. It is usual to include either of them in the analysis to avoid multicollinearity. This potential problem should be addressed by showing no correlation among variables (by checking multicollinearity) or excluding some factors correlated to other factors.

Author Response

                                                                                                     22nd February 2021

Dear reviewer for the International Journal of Environmental Research and Public Health

Regarding Manuscript ID ijerph-1097393 entitled “Tuberculosis and non-communicable disease multimorbidity: An analysis of the World Health Survey in 48 low- and middle-income countries”.

We would like to take this opportunity to express our thanks to you for the positive feedback and valuable comments for correction or modification.

We believe that these comments have resulted in an improved revised manuscript. The manuscript has been revised to address your comments. Please see your comments and our point-by-point responses to your comments, listed below:  

The authors aimed to explore the pattern and burden of multimorbidity in adults with TB and NCDs in 48 LMICs. Considering the established association between TB and NCDs and its scarce data in LMICs, this study gives an added value to the field. However, I would like to raise some concerns for publication.

Reply: We would like to thank the Reviewer for acknowledging the importance of our work and for offering valuable comments which have helped improve the quality of the manuscript considerably.

  1. Study limitations: The discussion section indicates several limitations. However, the most important limitation would be the study design. A cross-sectional study does not determine the causal relationship between factors. We do not know how these factors are correlated through this study design. This should be indicated in the discussion section. In addition, it would be better to mention how to have tried to address all other limitations in the manuscript.

Reply: We sincerely appreciate the reviewer’s comment on our limitation section and we totally agree with this suggestion. Therefore, additional sentences have been added following each limitation.

For example: ‘the utilization of self-reported measures for physical conditions and mental health diagnoses may have led to recall bias or underdiagnosis in our sample, especially in older subjects. On a similar note, the diagnosis of TB was also based on self-reported typical symptoms; therefore, there was also the possibility for misclassification. Whilst there are multiple other studies that have been published using this data in other populations [11, 17, 21], this consideration should be taken into account. Furthermore, distinction between drug-sensitive and MDR-TB could not be addressed. Therefore, future studies should include different types of TB and may wish to consider the inclusion of more objective diagnostic methods for physical and mental health conditions, such as patients’ hospital medical records.’

Also: ‘New surveys should be carried out and provide the most up-to-date evidence in order to inform the relationship between TB and NCDs in LMICs. Finally, our study did not include at-risk samples, such as homeless individuals or prisoners. The prevalence of chronic health conditions (e.g., infections, mental health problems) and subsequent all-cause mortality tends to be higher among marginalized populations [86-87]. Future research should thus consider these vulnerable and marginalized populations.’

The limitation of our cross-sectional study design has now been added in the limitation section:

‘First, this study is cross-sectional in nature, hence, the causality and temporality between TB and NCDs could not be inferred. Future research with a prospective cohort design will provide rigorous evidence of the relationship between TB and NCDs’.

  1. Study implications: What are the implications of the study? How can we apply these findings in the field? Which policy should be adopted to address these findings? The manuscript does not clearly mention these.

Reply: We thank the reviewer for this helpful comment. We have now included additional discussion on the implication of our findings, as follows:

Our findings have important implications in informing policy change and clinical practice. The key message is that when managing TB patients, due to the co-occurrence of TB with NCDs, some specific NCD multimorbidity clusters (e.g., angina and depression) deserve early detection and specific care. Frequent screening of NCDs should be provided in routine TB care. Furthermore, considering the burden and frequency of the co-occurrence of depression with NCDs [82] and with TB [59], our results suggest the need for a strategy for management that can take all these factors into consideration. Finally, continuous education regarding the co-occurrence of TB and specific NCDs or NCD multimorbidity for healthcare providers may lead to early detection of TB multimorbidity and better patient care.”

  1. Possible multicollinearity: In the multivariable logistic regression analysis, several variables are included for adjustment including income and wealth. In particular, I am concerned about possible multicollinearity between income level and education status. These two factors are usually correlated so that if two factors are included in a multivariable analysis, the result might be biased. It is usual to include either of them in the analysis to avoid multicollinearity. This potential problem should be addressed by showing no correlation among variables (by checking multicollinearity) or excluding some factors correlated to other factors.

Reply: Thank you for raising this issue. In order to assess the influence of multicolinearity, we calculated the variance inflation factor (VIF) value for each independent variable, and the results suggest that multicollinearity should not be a problem for this study. We have now included the following sentence in the methods section:

In order to assess the influence of multicollinearity, we calculated the variance inflation factor (VIF) value for each independent variable. The highest VIF was 1.43, which is much lower than the commonly used cut-off of 10 [57], indicating that multicollinearity was unlikely to be a problem in our analyses.’

Best Wishes,

Dr. Brendon Stubbs on behalf of all co-authors

Reviewer 2 Report

Why to NCD 's sample did not included patients with COPD and Bronchectasis?

Author Response

                                                                                                    22nd February 2021

Dear reviewer for the International Journal of Environmental Research and Public Health

Regarding Manuscript ID ijerph-1097393 entitled “Tuberculosis and non-communicable disease multimorbidity: An analysis of the World Health Survey in 48 low- and middle-income countries”.

We would like to take this opportunity to express our thanks to you for the positive feedback and valuable comment for correction or modification.

We believe that these comments have resulted in an improved revised manuscript. The manuscript has been revised to address your comment. Please see your comment and our point-by-point response to your comment, listed below:  

Why to NCD 's sample did not included patients with COPD and Bronchectasis?

Reply: We would thank the reviewer for reviewing our paper and for providing valuable suggestion which has helped improve the quality of the manuscript. The World Health Survey (WHS) only collected data on a total of 9 NCDs, and unfortunately COPD and bronchiectasis were not included. To make this clear, we have added additional sentences in the limitation section.

Our study included nine physical conditions but lacked information on some other important NCDs such as chronic obstructive pulmonary disease (COPD) and bronchiectasis. Therefore, the prevalence of multimorbidity could be higher if more conditions were included.’

Best Wishes,

Dr. Brendon Stubbs on behalf of all co-authors

Reviewer 3 Report

The others analyse the association between tuberculosis (TB) and non-communicable diseases (NCD) in low- and middle-income countries (LMIC) using data from a WHO survey from different countries. TB is associated with NCD either as a risk factor for Tb or because of TB. This is not surprising, but now it is described on a large scale. Focusing on TB alone might not be enough when trying to improve health for people living in LMIC. To me this seems of particular interest in times of the COVID-19 pandemic as we observe that TB might be neglected due to the focus on COVID-19.

The paper is well written. I have some concerns with two of the figures.

1) In figure 4, I guess percentage are presented. If I am right, probably this should be indicated.

2) Figure 5 is a bit confusing to me. It shows duration of living with a certain health impairment for those with and without TB in years. As the data is shown it looks like it is cumulative. Is this what the figure says? All impairments taken together add up to 5 years in those without TB and to 48 years including TB in those with TB. Duration of TB I would present separately. In addition, I propose to compare the duration for every NCD for example in columns.

Thank you for the chance to read this interesting paper.       

Author Response

                                                                                             22nd February 2021

Dear reviewer for the International Journal of Environmental Research and Public Health

Regarding Manuscript ID ijerph-1097393 entitled “Tuberculosis and non-communicable disease multimorbidity: An analysis of the World Health Survey in 48 low- and middle-income countries”.

We would like to take this opportunity to express our thanks to you for the positive feedback and valuable comments for correction or modification.

We believe that these comments have resulted in an improved revised manuscript. The manuscript has been revised to address your comments. Please see your comments and our point-by-point responses to your comments, listed below:  

The others analyse the association between tuberculosis (TB) and non-communicable diseases (NCD) in low- and middle-income countries (LMIC) using data from a WHO survey from different countries. TB is associated with NCD either as a risk factor for Tb or because of TB. This is not surprising, but now it is described on a large scale. Focusing on TB alone might not be enough when trying to improve health for people living in LMIC. To me this seems of particular interest in times of the COVID-19 pandemic as we observe that TB might be neglected due to the focus on COVID-19.

Reply: We would like to express our sincere gratitude to the Reviewer for acknowledging the importance of our work and for providing valuable comments which have helped improve the quality of our manuscript substantially.

The paper is well written. I have some concerns with two of the figures.

1) In figure 4, I guess percentage are presented. If I am right, probably this should be indicated.

Reply: We appreciate this comment. Figure 4 is a correlation matrix, so the numbers are not percentages but correlations. This is indicated in the title of the figure. However, numbers in figure 3 are in percentage. So, we have indicated this in the figure.

2) Figure 5 is a bit confusing to me. It shows duration of living with a certain health impairment for those with and without TB in years. As the data is shown it looks like it is cumulative. Is this what the figure says? All impairments taken together add up to 5 years in those without TB and to 48 years including TB in those with TB. Duration of TB I would present separately. In addition, I propose to compare the duration for every NCD for example in columns.

Thank you for the chance to read this interesting paper.

Reply: We are grateful for this comment. Figure 5 shows a cumulative total of Years Lived with Disability (YLD) in a group of people with or without TB. We estimated that by applying disability weights for respective conditions within the two cohorts. We have amended the caption to make this clear (the caption has been changed to ‘The cumulative total of Years Lived with Disability in a cohort of people with and without tuberculosis). Given that this is to highlight the difference in the cumulative total of YLD between the two groups, showing different conditions as separate columns may not portray this difference as clearly.

Best Wishes,

Dr. Brendon Stubbs on behalf of all co-authors

Round 2

Reviewer 1 Report

All points are well addressed. No further comments.